# Development of Temperature-Controlled Shear Tests to Reproduce White-Etching-Layer Formation in Pearlitic Rail Steel

**DOI:** 10.3390/ma15196590

**Published:** 2022-09-22

**Authors:** Léo Thiercelin, Sophie Cazottes, Aurélien Saulot, Frédéric Lebon, Florian Mercier, Christophe Le Bourlot, Sylvain Dancette, Damien Fabrègue

**Affiliations:** 1Arts et Métiers Institute of Technology, CNRS, Université de Lorraine, LEM3-UMR 7239, F-57078 Metz, France; 2Université de Lyon, INSA Lyon, CNRS UMR 5510, MATEIS, F-69621 Villeurbanne, France; 3Université de Lyon, INSA Lyon, CNRS UMR 5259 LaMCoS, F-69621 Villeurbanne, France; 4Aix Marseille Université, CNRS, Centrale Marseille, LMA UMR 7031, F-13453 Marseille, France

**Keywords:** hat-shaped specimen, shear stress, pearlitic steel, thermomechanical test, dynamic recovery, white etching layer

## Abstract

The formation of a white etching layer (WEL), a very hard and brittle phase on the rail surface, is associated with a progressive transformation of the pearlitic grain to very fragmented grains due to the cumulative passage of trains. Its formation is associated with a complex thermomechanical coupling. To predict the exact conditions of WEL formation, a thermomechanical model previously proposed by the authors needs to be validated. In this study, monotonic and cyclic shear tests using hat-shaped specimens were conducted in the temperature range of 20 °C to 400 °C to reproduce the WEL formation. The tests showed a strong sensitivity of the material to temperature, which does not necessarily favor WEL formation. For the monotonic tests, no WELs were produced; however, a localization of the plastic deformation was observed for tests performed at 200 °C and 300 °C. In this temperature range, the material was less ductile than at room temperature, leading to failure before WEL formation. At 400 °C, the material exhibited a much more ductile behavior, and nanograins close to WEL stages were visible. For the cyclic tests, a WEL zone was successfully reproduced at room temperature only and confirmed the effect of shear in WEL formation. The same cyclic tests conducted at 200 °C and 300 °C yielded results consistent with those of the monotonic tests; the deformation was much more localized and did not lead to WEL formation.

## 1. Introduction

The increase in rail traffic and cumulative tonnage over the past decades has led to an increase in rolling-contact fatigue defects on the rail surface [1]. The emergence of this increasing number of defects is generally associated with microstructural changes on the rail surface, such as severe plastic deformation and the formation of new, harder, and more brittle phases commonly known as brown and white etching layers (BEL and WEL, respectively) [2].

Understanding the kinetics of WEL formation is complex, as it depends on contact conditions such as thermomechanical, cyclic, multiaxial, and dynamic loading. Therefore, the thermomechanical path leading to WEL formation is not unique [3,4]. Nevertheless, microstructural characterization of the rail surface has provided evidence that the temperature and/or mechanical-stress field are the driving forces in WEL formation.

On the one hand, a purely thermal mechanism of WEL formation has been proposed by several authors [5,6,7,8,9,10]. Indeed, under high-braking conditions with high wheel sliding [3], the rail surface undergoes thermal cycles of heating with temperatures above 700 °C followed by rapid cooling that could progressively produce WEL spots. In addition, Nakkalil [11] explained that dynamic loading applied to the rail surface induces localized plasticity and would lead to the formation of multiple adiabatic shear bands in the rail surface. These shear bands would then accumulate to form a homogeneous WEL, as also explained by Baumann [12]. This process would explain the presence of martensite and residual austenite within the thermal WEL [13].

On the other hand, under conventional traffic conditions on aligned tracks, the relative sliding ratio between the wheel and rail is generally limited to 2% by anti-skid devices [14], which limits the contact temperature to only several hundred degrees [15,16]. Therefore, wheel–rail contact tests performed at a sliding ratio above 2% are not representative of the wheel–rail contact and highlight the thermal WEL formation [17,18]. WELs observed in areas subjected to low temperature rise are composed of highly deformed grains associated with an accumulation of plastic deformation [3,4,19,20]. The severe plastic deformation (SPD) progressively transforms pearlitic grains of several dozen of micrometers into nanograins of dislocated ferrite supersaturated in carbon [2,21]. In fact, the increase of the dislocation density would facilitate the fragmentation of the grains [22,23,24,25,26,27,28], increase the kinetics of cementite dissolution [29,30], and favor the carbon-atom mobility [31]. In addition, such a level of deformation would be linked to the intrinsic conditions of the wheel–rail contact that combines high compressive stresses with high shear stresses. Under these severe conditions, the material could locally reach a level of severe deformation that would be analogous to the severely deformed specimens in the case of cold drawing [32,33,34] or SPD experiments [23,35,36]. Indeed, high-pressure torsion tests [36] performed for pearlitic steels at room temperature confirmed the mechanically induced WEL formation under hydrostatic pressure and shear coupling.

These two scenarios cannot really be distinguished, and the hypothesis of a thermomechanical coupling is the most probable to explain and predict WEL formation [2,37]. A macroscopic model developed by Antoni et al. [38] was then proposed to simulate the mechanical-stress effect on WEL formation. In this model, a coupling between the hydrostatic pressure and temperature was proposed. Nevertheless, this model does not take into account the shearing of the grains at the extreme surface of the rails. To overcome these limits, Thiercelin et al. [39] improved this model by adding a shear contribution to the WEL formation criterion. The WEL formation would result from a coupling between the hydrostatic pressure and the shear stress, which would be enhanced by the temperature. The tendencies of WEL formation kinetics have already been confirmed in the literature [36,40,41,42]; however, the model needs to be more accurately experimentally identified. For this purpose, this coupling has been de-correlated to separately quantify the effects of temperature, hydrostatic pressure, and shear.

First, Merino [43], Lafilé [44], and Thiercelin [45] successfully reproduced the kinetics of WEL formation under representative wheel–rail contact conditions at 1/15th scale. These tests demonstrated the predominantly mechanical formation of the WEL, as the slip level was less than 2%. Moreover, the combined effect of contact pressure and sliding was confirmed, validating one part of the model proposed by Thiercelin et al. [39], i.e., the pressure–shear coupling.

Second, temperature-controlled shear tests have demonstrated the formation of adiabatic shear bands in steels [46,47]. In addition, tests performed by Lins et al. [48] on low-carbon interstitial-free steels (IF steel) revealed shear-band formation kinetics, called “progressive subgrain misorientation”, which is similar to the mechanically induced WEL kinetics. Such tests conducted on pearlitic steels are still lacking and would constitute a major breakthrough for the identification of the model proposed by Thiercelin et al. and more generally for the formation of shear bands in pearlitic steels.

The objective of the current study was to reproduce shear tests under controlled temperatures representative of wheel–rail contact. Monotonic and cyclic shear tests were conducted for different temperature ranges (from 20 °C to 400 °C). Optical and scanning microscopy observations were performed to characterize each test condition. First, static monotonic tests with microstructure analysis are presented for each temperature tested (20 °C, 200 °C, 300 °C, and 400 °C). Next, the results of the cyclic tests are presented for each tested temperature. Finally, the effect of temperature and loading path are discussed with respect to the microstructural evolution eventually leading to WEL formation.

## 2. Materials and Methods

### 2.1. Material

The material studied was a pearlitic rail steel grade R260 used in most of the French railway network. The chemical composition of the material is given in Table 1.

In its as-received state, the microstructure consists of pearlitic grains with a diameter measured to be approximately 9.6 ± 8.1 μm. The grains initially have a rather globular shape with an aspect ratio of 2.0 ± 0.9 and are weakly disoriented. There are initially 30% low-angle grain boundaries (LAGBs), a majority of medium-angle grain boundaries (MAGBs) (40%), and 30% high-angle grain boundaries (HAGBs) (Figure 1a). The group of grains having a similar crystallographic orientation constitutes a pearlitic colony whose average size is generally several tens of micrometers [50] (Figure 1b).

### 2.2. Thermomechanical Test Bench

Thermomechanical tests were performed using the Gleeble 3800 device, a test bench that can simulate a complex thermomechanical loading path by performing mechanical cycles and thermal cycles simultaneously (Figure 2). The Gleeble thermomechanical test bench can perform compression tests at a rate up to 2000 mm/s. The measurement and control of the strain was performed by the displacement of the mobile die previously calibrated. In addition, the specimen temperature was measured with type K thermocouple welded on the sample. The specimen was heated by conduction, and the temperature was continuously monitored with a control loop adapting the amount of current in the sample in order to get the desired thermal cycle. For more information, please refer to the Gleeble company website [51].

### 2.3. Hat-Shaped Specimen

The shear tests under controlled temperature were performed with the help of a hat-shaped specimen, which, under the effect of a compression, generated a localized shear in a crown of width *L* = 2 mm and thickness *d* = 0.2 mm (Figure 3). The thickness *d* corresponded to the size of the deformation gradient observed at the rail surface. The other dimensions were chosen from previous work on similar alloys [52,53] and are summarized in Table 2.

To analyze the experimental data, some theoretical assumptions about the stress and strain field in the sheared zone are required. The hat specimens are assumed to undergo pure shear, which is confined to the theoretical shear zone (green area in Figure 3). The stress and strain tensors are then expressed as follows:(1)$$\sigma =\left(\begin{array}{ccc} 0 & \tau & 0\\ \tau & 0 & 0\\ 0 & 0 & 0 \end{array}\right)\;\mathrm{with}\;\tau =\frac{F}{\pi L(r_{i}+r_{e})}$$
(2)$$\varepsilon =\left(\begin{array}{ccc} 0 & \gamma /2 & 0\\ \gamma /2 & 0 & 0\\ 0 & 0 & 0 \end{array}\right)\;\mathrm{with}\;\gamma =\tan\left(\alpha \right)=\frac{u}{d}$$
with τ representing the shear stress induced in the sheared zone, which depends on *F*, the applied compressive load; *L*, the length of the sheared zone; and $r_{i}$ and $r_{e}$, the inner and outer radius of the specimen, respectively (Figure 3). In addition, γ is the shear strain, which depends on *u*, the displacement in the compression direction, and *d*, the theoretical thickness of the sheared zone.

It must be pointed out that the heterogeneous shape of the specimen induces temperature heterogeneity in the specimen. Preliminary heat-conduction tests showed that there was a difference of approximately 10 °C between the lower and upper part of the specimen. Nevertheless, the current study was focused on the effect of the temperature in the shear zone; therefore, only one thermosensor was welded in the vicinity of the shear stress (Figure 4), and the temperature was considered uniform in the specimen during all the mechanical tests.

### 2.4. Microstructural Characterization

Optical microscopy characterization was performed after mechanical polishing down to 1 μm and subsequent Nital etching to observe cementite lamellae and the possible WEL.

The samples were also prepared for observation in secondary electron (SE) mode and electron backscatter diffraction (EBSD) mode in the scanning electron microscope. For the EBSD mode, the samples were mechanically polished down to 1 μm using a conventional grinding machine. The final preparation step consisted of vibratory polishing for approximately 1 h using a colloidal silica suspension (Struers OP-S) with a grain size of 0.05 μm. For some cross-section samples, it was possible to perform an additional ionic polishing step with a GATAN ILLION 2 device with an accelerating voltage of 4 kV. The observations were performed using a FEG Zeiss Supra 55VP microscope equipped with an Oxford EBSD Symmetry detector. The accelerating voltage was set to 12 or 15 kV, depending on the size of the microstructural elements to be analyzed. The EBSD data treatment was performed using Atex software [54]. For the data treatment of the EBSD maps, the disorientation angle selected for grain detection was set to 5°, and only grains with at least 10 pixels were considered, as the detection is not reliable below this level. The statistical distributions of the grain size and the aspect ratio in each EBSD map were characterized from mean values followed by the the standard deviation noted “±”. Figure 5 indicates the color code used to define the crystallographic textures of all the inverse pole figures (IPF) presented in this study.

The sheared areas of each specimen were then analyzed after having been axially cut and prepared as previously described. It is worth noting that some sheared zones were analyzed after the failure of the specimen and some were analyzed before. Depending on the final state of the specimen, an associated scheme is included to facilitate understanding of the microstructural analyses (Figure 6).

The sheared zones were systematically compared to the stages of microstructural evolution leading to the WEL formation observed on the rail surface. Based on a mechanism proposed by several authors [25,28] and the formalism initially proposed by Thiercelin et al. [45], three microstructural indicators were used to quantitatively describe the progressive evolution of the microstructure: mean grain size, aspect ratio, and grain-boundary disorientation. For the latter, three intervals of grain disorientation were considered:LAGBs for angles between 5° and 15°,MAGBs for angles from 15° to 40°,HAGBs for angles above 40°.

Figure 7 is a longitudinal cross-section of worn pearlitic rail with evidence of a microstructural gradient from the pearlitic stage to the WEL stage over approximately 60 μm from the contact surface. The microstructural gradient analysis provides insights into the kinetics of WEL transformation. This gradient can be divided into several successive stages of transformation located at different depths. First, the grains begin to fiber and remain mainly unfragmented (stage 2) from 40-μm depth. The average grain size starts to decrease, the aspect ratio is very high, and the grain disorientation remains quite low. Next, between 20- and 40-μm depth, the grains fragment but still keep a rather elongated shape (stage 3). At 10-μm depth, there is a fine area of some micrometers composed of strongly disoriented nanograins. This is the non-fibrous and nanostructured stage (stage 4). Finally, in the first 10 μm from the contact surface, the grains appear white after Nital etching, which corresponds to a WEL spot. At this stage, the nanograins are spherical and much more disoriented than in the previous stages. This stage is considered the final stage in WEL formation (stage 5). The stages of transformation leading to WEL formation and the associated microstructural indicators proposed by [45] are presented in Table 3.

## 3. Results

### 3.1. Monotonic Tests

As the temperature in the wheel–rail contact does not exceed several hundred degrees, testing temperatures of less than 400 °C were selected. First, monotonic shear tests were performed at four temperatures until failure (20 °C, 200 °C, 300 °C, and 400 °C) at a quasi-static loading rate ($\dot{\gamma }$ = 0.5 $s^{-1}$). The conditions of the monotonic tests are given in Table 4. A macroscopic analysis of the experimental curves combined with microstructural characterization of each condition tested was conducted.

#### 3.1.1. Macroscopic Analysis

The macroscopic behavior of the material was studied by plotting the stress–strain curve (Figure 8) obtained with the stress and the associated shear strain as theoretically observed by the material in the sheared zone (Equations (Equation 1) and (Equation 2)). For each temperature, at least two experiments were performed and showed similar stress–strain response, confirming the repeatability of the experiments.

For temperatures below 300 °C, the ductility of the material decreased with temperature. The failure strain was on the order of 5.8 at room temperature, whereas it was 5.1 at 200° and 4.9 at 300 °C. Moreover, in this temperature range, the stress–strain curves at 20 °C and 300 °C followed the same trend, differing from that at 200 °C. Indeed, for the latter, the material became significantly less resistant than at 20 °C and 300 °C as its apparent yield stress (stress beyond which the behavior becomes non-linear) and its critical shear stress before failure were much lower. The observations at these three temperatures reveal three different behaviors, which do not allow conclusions to be drawn on the trend of behavior in this temperature range.

In addition, the curves at 200 °C and 300 °C exhibit serrations characterized by microstress drops during monotonic loading. This mechanism can likely be attributed to the Portevin–Le Chatelier effect (PLC) already encountered for pearlitic steels in this temperature range [56,57,58]. This mechanism, also known as dynamic strain aging (DSA), corresponds to an instability of the plastic flow in metals when dislocations interact with atoms in solid solutions. During plastic deformation, the dislocations are blocked by the atoms in solid solutions until a critical force is reached. Subsequently, the stress falls until the next obstacle, explaining the serrations.

It is, however, quite surprising that similar serrations were observed regardless of the temperature. Indeed, previous studies have shown that the PLC is favored at higher temperatures in pearlitic steels. Therefore, we consider that the serrations observed could also be experimental artifacts.

For a temperature of 400 °C, the material response differed completely from that at the previous temperatures tested. It became softer and more ductile compared with the material at the other temperatures, with a failure strain above 7. It must be pointed out that for this experiment, the specimen did not fail. Moreover, the apparent yield strength estimated at 300 MPa drastically decreased compared with that at 20 °C (approximately 450 MPa).

The following section will focus on the microstructural characterization of the shear zone performed at the end of the monotonic test for each temperature.

#### 3.1.2. Microstructural Characterization

Figure 9 presents optical and EBSD micrographs of the sheared area obtained at the end of the test for each temperature. Similar to the macroscopic analyses, the microstructural behavior in the range between 20 °C and 300 °C differed slightly from that at 400 °C. The observations are thus given separately.

First, between 20 °C and 300 °C, a microstructural gradient similar to that of the rail surface was observed. At room temperature, the microstructural gradient extended over a hundred micrometers with grains that gradually transformed from the broken surface, in a fragmented and unfibered state (stage 4), to a fibrous stage at a depth of 100 μm (stage 2). In the intermediate area, between 30 and 60 μm, a mixture of grains making the transition between stage 3 and 4 was observed. These optical observations are confirmed by the EBSD map within this area, where the grain have a relatively globular shape with an aspect ratio of 2 ± 1 and are submicrometric in size (0.5 ± 0.2 μm). In addition, their disorientation is rather high as the proportion of MAGBs and HAGBs is predominant at 39% and 33%, respectively. The evaluation of all these indicators lead to the conclusion that the grains in the area studied by EBSD are in a transition stage between stages 3 and 4 (Table 3) as seen in Table 5.

The same test performed at 200 °C revealed a similar microstructural gradient as that at room temperature. The sheared zone consisted of sheared grains that gradually became less fragmented and fibered to a depth of 60 μm. This sheared zone appears more confined than that for the test performed at 20 °C, which was approximately 100 μm thick. The effect of temperature on the strain localization appears to confirm the results at 300 °C, where the grains were slowly sheared. The analysis of the stress–strain curves combined with the microstructural gradient indicate that the plastic deformation was much less severe at 200 °C and 300 °C, where it is very confined.

As noted above, the material was much more ductile at 400 °C and did not fail. The micrograph shows the microstructural gradient of the specimen strained until a shear strain of 6.9. No WEL spots were observed; however, the grains were highly fibered over a large thickness of 200 μm, which corresponds to the theoretical sheared zone *d* (Figure 3). A main crack that initiated at the corner of the specimen propagated along the interface between the fibered structure and the as-received pearlitic grains. We also denote the presence of many secondary cracks within the sheared zone.

The EBSD map of the area close to the crack tip reveals a strong grain heterogeneity in the sheared zone. The grains on either side of the crack were deformed and reoriented in the shear direction without being fragmented (stage 2). Close to the most sheared zone, the grains were fibered and fragmented (stage 3). At the tip of the crack, the grains were not fibered and were very fragmented and randomly oriented, corresponding to a critical grain state close to the WEL state (stage 4). In addition, a network of secondary cracks in the presence of grains at the same critical stage (stage 4) nearby were observed.

### 3.2. Cyclic Tests

#### 3.2.1. Macroscopic Analysis

Cyclic shear tests were then conducted for different temperatures to simulate the thermomechanical loading path undergone by the rail. Because WEL formation is related to the cementite dissolution by mechanical stresses and, in particular, by dislocations, a maximal cyclic stress slightly higher than the apparent elastic shear stress determined with the monotonic stress–strain curves was applied (Figure 8). In addition, to maintain contact and ensure good thermal conduction, a minimum cyclic shear stress of 100 MPa was imposed, which corresponds to the minimum contact force of 5 kN.

A maximum stress of 560 MPa was considered for 20 °C and 300 °C. The specimen failure occurred after 600 cycles compared to 1400 cycles at 300 °C.

For a temperature of 200 °C, only a few cycles would have led to the failure of the specimen. At this temperature, a lower maximum stress of 500 MPa was thus considered. A test of 10,000 cycles was conducted without breaking the specimen. The thermomechanical conditions for the cyclic tests are summarized in Table 6.

These preliminary results highlight the significant effect of temperature on the fatigue limit of the material. In a similar manner as for the monotonic case, a different behavior was observed for each temperature. Indeed, between 20 °C and 200 °C, the fatigue limit of the material decreased as the temperature increased. In contrast, at 300 °C, the fatigue limit unexpectedly became greater than at 20 °C. Cyclic tests before specimen failure were conducted for all three temperatures. The following section will present the microstructural observations of the three tests before failure.

#### 3.2.2. Microstructural Characterization

Figure 10 presents optical micrographs after Nital etching of the shear zone at the end of each cyclic test stopped before the specimen broke. Regardless of the test temperature, a crack usually started in the corner of the cap and propagated over a few hundred micrometers.

For the test at 20 °C, the grains downstream of the crack were sheared but appeared slightly fragmented (stage 2). Moreover, on both sides of the crack, a transformation gradient up to the WEL was visible (see the optical zoom in this area). It can be assumed that cracks might have propagated due to incompatibility of the microstructure.

An EBSD map of the region around the crack (red box in the micrograph at room temperature of Figure 10) is presented to measure the microstructural indicators in this area (Figure 11). This map is divided into four distinct areas as follows:A fibered and fragmented zone in the lower part of the image (zone A);A non-indexed zone in the upper part of the crack, which corresponds to the white zone observed optically (zone B);A very fragmented zone without fibration above the very poorly indexed zone (zone C);A transition zone where the material fibered and flowed until it fragmented (zone D).

Region A, the lower part of the crack, was composed of fibered and elongated grains with an average size of 0.4 ± 0.2 μm and an aspect ratio of 2 ± 1, on average. In this region, there is a majority proportion of low disoriented grains (45%) and fairly close proportions of weakly and very disoriented grains (38% and 18%, respectively). All of these indicators are consistent with the identification of stage 3 in zone A.

Region B, corresponding to a WEL optically, was unindexed in EBSD, suggesting a significant grain-size reduction that is smaller than the indexation stepsize of 0.06 μm and a high level of grain disorientation.

Region C, which was optically dark (Figure 10, T = 20 °C), did not show evidence of any apparent grain fibering. This zone consisted of spherically shaped nanograins (aspect ratio of 1.6 ± 0.5), with a fairly uniform size (0.3 ± 0.1 μm). The grains had no particular crystallographic orientation and were much more disoriented than those in region A. Indeed, weakly disoriented grains occupied only 12% compared to 45% in zone A. There was then a large majority of large disorientation angles equally distributed between the moderately disoriented and highly disoriented grains (43% and 45%, respectively). All these observations are indicative of stage 4 in this region.

The set of indicator values in zones A and C of Figure 11 are summarized in Table 7.

The micrographs at 200 °C and 300 °C exhibit the same microstructure (Figure 10). Contrary to the test performed at room temperature, the crack, being initiated at the corner, propagated normally to the shear direction. The grains were slowly deformed downstream to the crack, and no WEL was observed.

## 4. Discussion

### 4.1. Effect of the Thermomechanical Path on WEL Formation Kinetics

The monotonic tests conducted at the four temperatures from 20 °C to 400 °C did not lead to WEL formation. The monotonic tests revealed a strongly non-linear behavior of the material with temperature. Indeed, at 200 °C, the material was softer and more brittle than at room temperature. At 300 °C, behavior similar to that at 20 °C was observed, with the material being more resistant and ductile. At 400 °C, the stress–strain curve indicated a much more ductile behavior, which can be explained by the higher ductility of the grains [56]. WEL formation was never observed after monotonic tests. However, the analysis of their microstructure indicates that the temperature does not seem to favor the deformation/fragmentation required for WEL formation. Indeed, an analysis of the microstructures using the WEL formation indicators defined in Table 3 reveals that the most advanced stages were found for the RT sample. Table 8 summarizes the different microstructures obtained at the end of the monotonic tests.

Regarding the cyclic tests, a WEL was observed at 20 °C. However, at 200 °C and 300 °C, little deformation or transformation of the grains was observed. Table 9 summarizes the different microstructures obtained at the end of the cyclic tests.

It appears that WEL formation is possible under the effect of shear at room temperature via a mechanism of grain fragmentation and transformation for a sufficient level of deformation, as presented by Thiercelin [45]. Indeed, the tests performed at higher temperatures (200 °C and 300 °C) resulted in less deformation of the microstructure for both the static and cyclic tests compared with that at room temperature. Localization of the stress was observed, which is likely related to the geometry of the specimens, which would crack before forming a WEL.

In addition, in the 200–300 °C temperature range, the kinetics of WEL formation could be inhibited by other microstructural mechanisms, such as dynamic recovery. The latter would enhance the annihilation of dislocations and therefore enhance the deformability of the microstructure.

Another physical mechanism could concern the cementite precipitation beginning at 300 °C [36], which would counterbalance its dissolution under the effect of mechanical stresses. Finally, the more ductile behavior at 400 °C can be attributed to a mechanism of deformation of each phase of the grains (ferrite and cementite) without fragmentation [57].

### 4.2. Effect of Temperature on Fatigue Strength

In this study, the effect of contact temperature on the fatigue strength of the rails was studied in order to anticipate the failure of the rails. At room temperature, the material was much more ductile, and failure occurred after WEL formation. For higher temperatures (200 °C and 300 °C), the material was less ductile and the failure occurred after a more localized deformation. Finally, the monotonic test conducted at 400 °C revealed crack initiation in highly nanostructured zones.

The failure mechanism clearly differs with temperature, and the deformation of the microstructure at stages less advanced than WEL formation can already be critical for the material depending on the contact temperature.

Nevertheless, the stage of transformation of the microstructure combined with the contact temperature is a good indicator of the probability of material failure. The WEL stage is the most critical. This assumption can be related to the studies of Saxena et al. [59], where the toughness of WELs and work-hardened zones of rail steels were studied. The authors concluded that in the case of wheel–rail contact, the toughness of the material is inversely proportional to its hardness. As hardness is linked to the dislocation density, grain size, and carbon content in solid solution [32,60,61], the toughness would then be directly dependent on the evolution stages (Table 3).

### 4.3. Thermomechanical Model and Wheel-Rail Contact Conditions

The role of temperature in the kinetics of WEL formation is contestable, as temperatures above 200 °C will activate other microstructural mechanisms that inhibit grain fragmentation–transformation kinetics. The model proposed by Thiercelin [39] would then be invalidated for temperatures of 200 °C, 300 °C, and 400 °C. Future tests at lower temperatures (below 200 °C) will have to be conducted to determine whether low-temperature elevations could still favor the WEL formation mechanism, as noted by Newcombs et al. [19]. In addition, for temperatures close to the austenitization temperature, the stresses could facilitate WEL formation [1,62,63].

The second point concerns a noteworthy difference in the thermomechanical path of the tests and the reality of the wheel–rail contact. Indeed, contrary to the contact conditions, the temperature was kept constant during the entire duration of the applied cyclic mechanical loading. In reality, the cyclic thermal load is more complex than in those tests and could modify the kinetics of the microstructural transformations. Therefore, the WEL formation criteria would then depend explicitly on the temperature evolutions with time (heating and cooling rates) and on the accumulation process of deformation/temperature/shear rather than on the temperature itself.

Finally, the interactions of the different scales of the wheel–rail contact (train, wheel–rail interaction, and the effect of wear particles) lead to a variability of the thermomechanical field applied to the rail surface. The tests performed in this study showed a strongly non-linear behavior of the material, which was explained by the activation of various microstructural mechanisms, such as dynamic recovery, cementite precipitation and or dissolution, grain fragmentation, phase transformation, and/or deformation. Depending on the temperature, the material will have different cyclic responses, which may lead to the development of rolling-contact fatigue defects or contribute to the wear rate if the contact temperature is sufficiently high, as observed during bi-disc tests for temperatures above 300 °C [64].

## 5. Conclusions

Monotonic and cyclic shear tests under controlled temperatures were performed using hat-shaped specimens and the Gleeble thermomechanical simulator in an attempt to reproduce the WEL formation induced at the rail surface.

The monotonic tests conducted at four temperatures (20 °C, 200 °C, 300 °C, and 400 °C) did not lead to WEL formation but resulted in grain-transformation stages close to WEL formation. The cyclic tests conducted at 20 °C, 200 °C, and 300 °C in a plastic regime confirmed the monotonic trends and resulted in WEL formation at room temperature only. The effect of medium temperature appears to be unfavorable for WEL formation by fragmentation-transformation of grains as it activates other mechanisms, such as dynamic recovery or cementite precipitation.

The thermal coupling part of the model proposed by Thiercelin et al. [39] was then invalidated for temperatures between 200 °C and 400 °C; however, the formation of WEL by pure shear was confirmed. Further shear tests at temperatures between 20 °C and 200 °C must be considered to validate the model at low temperature.

The residual stresses in the sheared zone will be further estimated to improve the constitutive model of WEL formation. Moreover, these measurements will be compared to the residual stresses measured on the real rail surface in presence of WEL [6,7].

The tests confirmed that the evolution stages leading to WEL formation could reflect a probabilistic criterion of crack initiation. The probability of cracking of the material increases inversely with decreasing grain size, which is related to the stage of evolution of the microstructure. In the railroad, the crack initiation could then be explained by an advanced stage of evolution of the microstructure (stage 4 or 5).

These tests showed limitations due to premature cracking, which could be overcome by performing identical tests under hydrostatic pressure to limit the damage [65,66] and to be more representative of the multiaxial stresses undergone by the rail. Moreover, the achievement of thermal cycles simultaneously with the mechanical cycles constitutes future perspectives to simulate more faithfully the thermomechanical path undergone by the rail.

## Figures and Tables

**Figure 1 materials-15-06590-f001:**
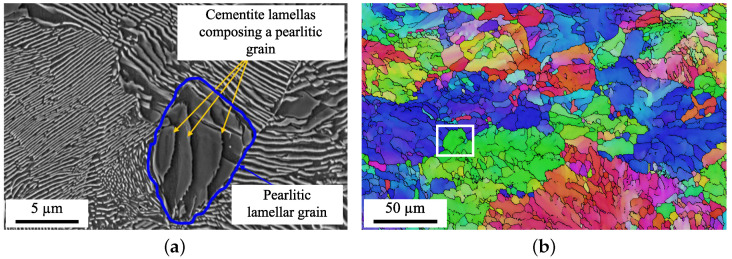
(**a**) SEM microstructure of the as-received structure and (**b**) EBSD IPFZ map of the as-received microstructure.

**Figure 2 materials-15-06590-f002:**
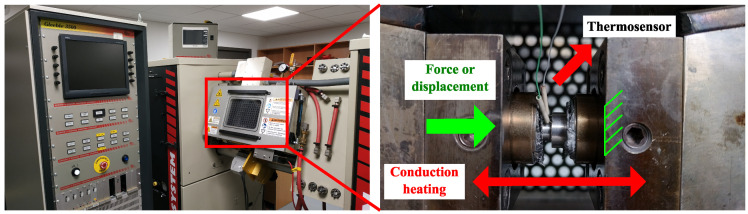
Gleeble 3800 test bench with the positioning of the hat-test specimen: the force (or displacement) is applied by a mobile die, and the temperature is induced by thermal conduction and is controlled with a type K thermocouple. For more information please see [51].

**Figure 3 materials-15-06590-f003:**
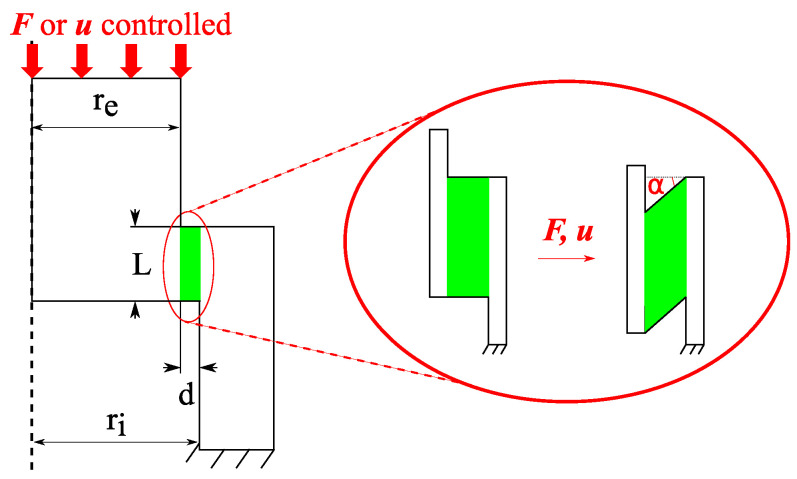
Scheme of an axisymmetric section of a hat-shaped specimen subjected to a compressive load with $r_{i}$ and $r_{e}$ representing the internal and external diameters of the specimen, respectively, and $d=r_{i}-r_{e}$ and *L* representing the thickness and width of the sheared zone, respectively. The sheared zone is then a crown of length *L* and thickness *d*.

**Figure 4 materials-15-06590-f004:**
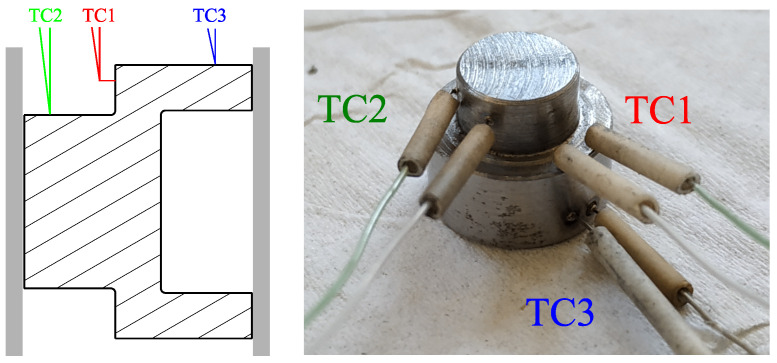
Welding of the thermosensor. The control of the temperature was achieved with the sensor labelled “TC1”.

**Figure 5 materials-15-06590-f005:**
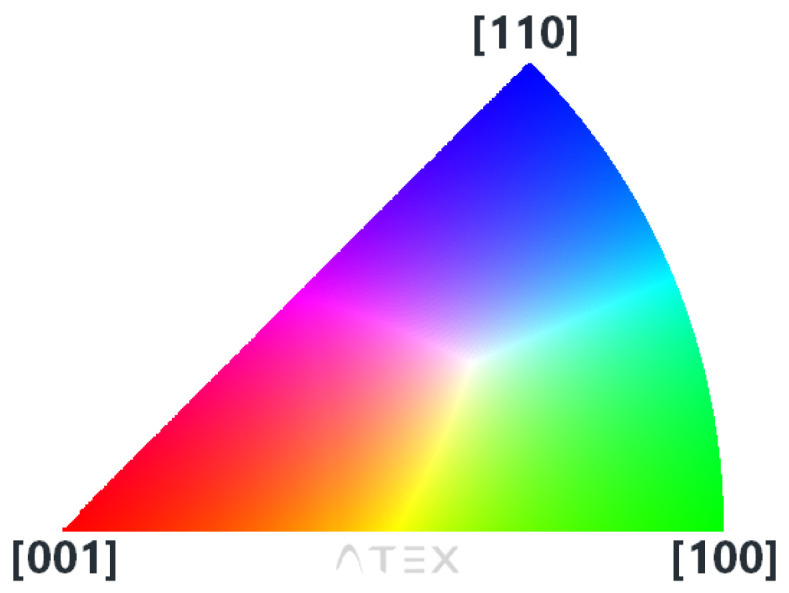
Inverse pole figure color coding of orientation maps presented in this study.

**Figure 6 materials-15-06590-f006:**
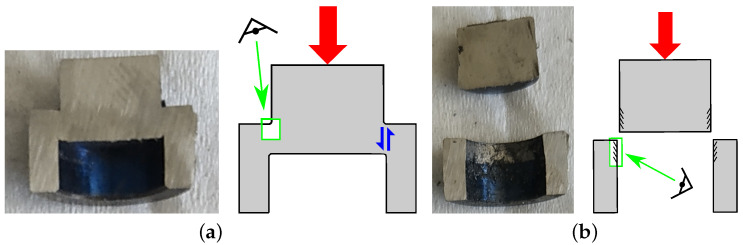
Axial cut of hat-shaped specimen: (**a**) case without failure and (**b**) case after failure.

**Figure 7 materials-15-06590-f007:**
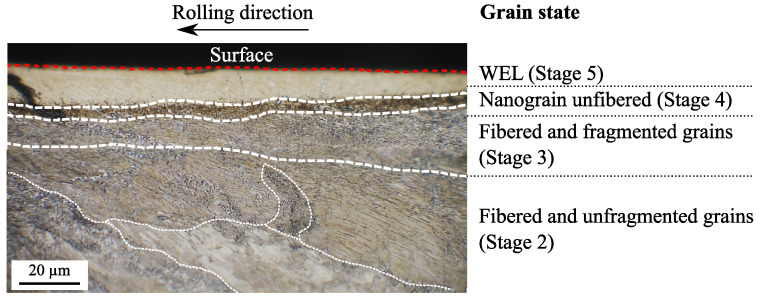
Longitudinal cross section of a rail extracted from region with a high WEL frequency [55]; the microstructural gradient consists of four areas representing the differents evolution stages leading to WEL formation. Stage 2 is the fibrous state of the grains without fragmentation; stage 3 is a state where the grains are still fibrous but fragmented; stage 4 is a state where the grains have no particular orientation and are very fragmented; and stage 5 is the state that appears white in optical microscopy after Nital etching.

**Figure 8 materials-15-06590-f008:**
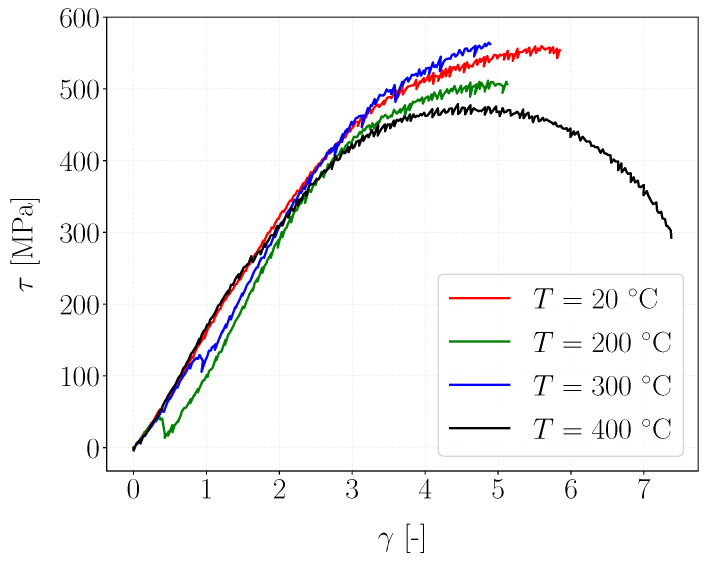
Shear stress τ vs. shear strain γ curve using (Equation 1) and (Equation 2).

**Figure 9 materials-15-06590-f009:**
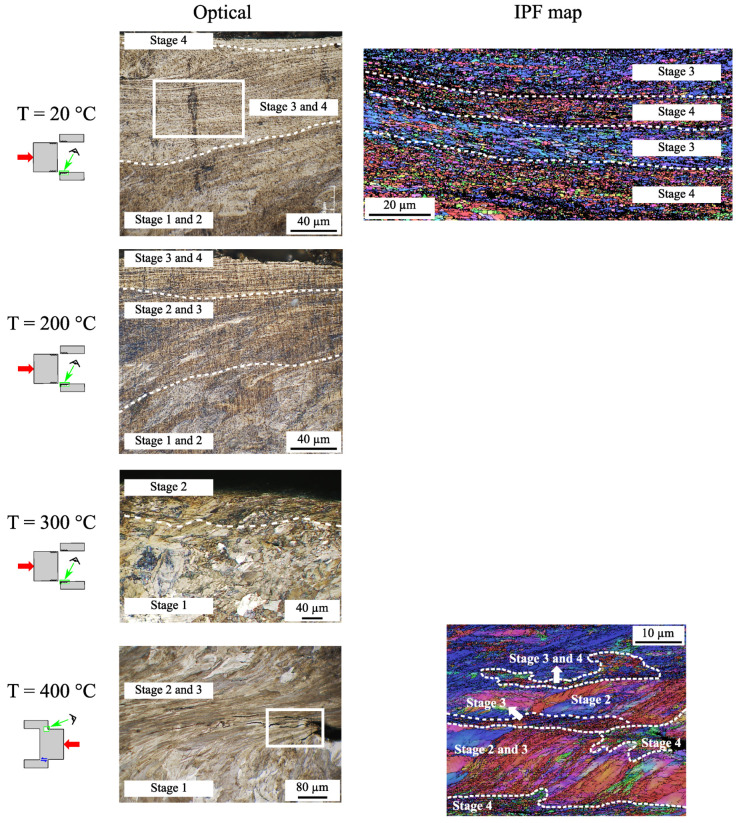
Effect of temperature on the microstructural gradient in the sheared zone of the specimen at the end of the monotonic tests. The specimen failed at the tests performed at T = 20 °C, 200 °C, and 300 °C but did not fail at 400 °C. IPF maps were constructed for the tests performed at room temperature (step size = 0.08 μm) and 400 °C (step size = 0.1 μm).

**Figure 10 materials-15-06590-f010:**
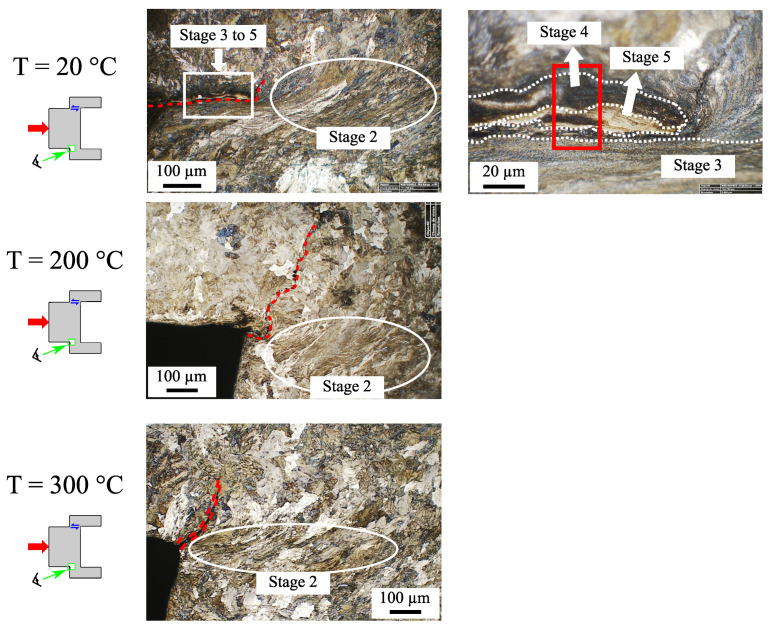
Effect of temperature on the microstructural gradient in the sheared zone of the specimen at the end of the cyclic tests. The mechanical loading and the number of cycles at each temperature are shown in Table 6. The red dotted line denotes the start of cracking in the corner of the specimen. The red box corresponds to the area investigated by EBSD in Figure 11.

**Figure 11 materials-15-06590-f011:**
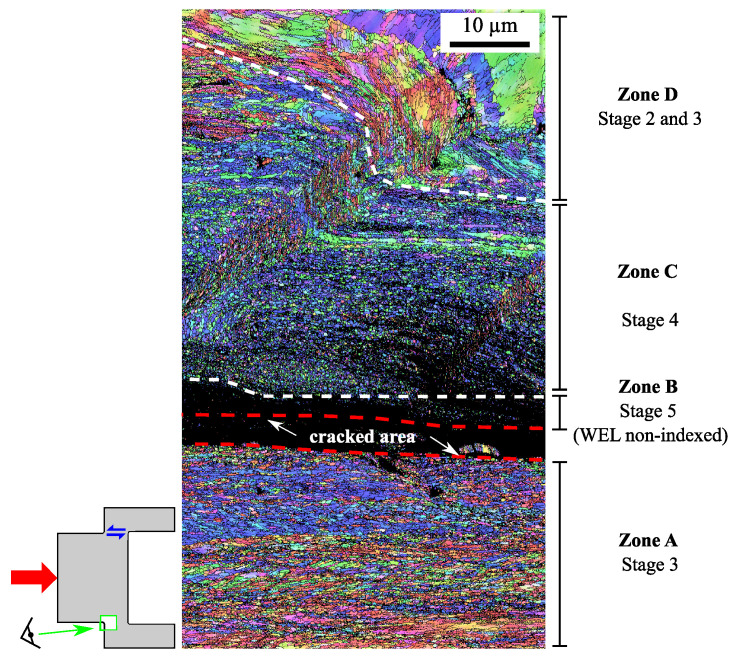
T = 20 °C − $N_{\mathrm{cycle}}$ = 500 with $\tau_{\min}/\tau_{\max}$ = 100/560 MPa, $\gamma_{\mathrm{final}}=3.5$. IPF X map of the microstructural gradient of the cracked zone that corresponds to the white box in Figure 10 for the case at room temperature, stepsize = 0.06 μm. This region is divided into four areas: zone A is composed of fibered and fragmented grains (stage 3). Zone B is a WEL area with very small grains and disoriented grains. Zone C is composed of very small and disoriented grains (stage 4), whereas zone D is a transition area with grains that begin to fiber and fragment themselves (stage 2 and 3).

**Table 1 materials-15-06590-t001:** Chemical composition of R260 pearlitic steel (weight%) [49].

C	Si	Mn	P	S	Cr	Al
0.62	0.80	0.15–0.58	0.70–1.20	<0.025	<0.15	<0.004

**Table 2 materials-15-06590-t002:** Hat-shaped specimen dimensions.

$r_{i}$ (mm)	$r_{e}$ (mm)	$d=r_{i}-r_{e}$ (mm)	*L* (mm)
4	3.8	0.2	2

**Table 3 materials-15-06590-t003:** Evolution of indicators and the stages of evolution from the pearlitic state to the final WEL state according to [45].

Stage	Grain State	Grain Size (μm)	Aspect Ratio (−)	LAGB-MAGB-HAGB (%)
1	As-received pearlitic	9.6±8.1	2.0±0.9	30-41-29
2	Fibered and unfragmented	0.6±0.3	4.1±2.7	20-48-32
3	Fibered and highly fragmented	0.6±0.4	2.8±1.4	19-50-31
4	Unfibered and nanostructured	0.2±0.1	1.7±0.7	9-48-43
5	WEL	0.2±0.1	1.6±0.5	20-18-61

**Table 4 materials-15-06590-t004:** Monotonic test conditions until failure.

Temperature (°C)	$\dot{\gamma }$ ($s^{-1}$)	Number of Tests
20	0.5	2
200	0.5	2
300	0.5	1
400	0.5	2

**Table 5 materials-15-06590-t005:** Evaluation of microstructural indicators for the EBSD map at room temperature.

Grain Size (μm)	Aspect Ratio	LAGB-MAGB-HAGB (%)	Stage Estimated
0.5 ± 0.2	1.9 ± 0.7	28-39-33	3 and 4

**Table 6 materials-15-06590-t006:** Cyclic tests conditions.

Temperature (°C)	$\tau_{\min}$ (MPa)	$\tau_{\max}$ (MPa)	Number of Cycles
20	100	560	500 and 600 (failure)
200	100	500	10,000
300	100	560	1000 and 1400 (failure)

**Table 7 materials-15-06590-t007:** Summary of measurements of indicators for the EBSD map in Figure 11.

Zone	Grain Size (μm)	Aspect Ratio	LAGB-MAGB-HAGB (%)
A	0.4 ± 0.2	2.0 ± 1.0	45-37-18
C	0.3 ± 0.1	1.6 ± 0.5	12-43-45

**Table 8 materials-15-06590-t008:** Synthesis for the monotonic tests.

Temperature (°C)	$\gamma_{\mathrm{failure}}$ (−)	Presence of WEL	Final Stage
20	5.3	no	4
200	4.2	no	4
300	4.4	no	2
400	6.9	no	4

**Table 9 materials-15-06590-t009:** Synthesis for the cyclic shear tests. The same maximal stress of 560 MPa was considered for 20 °C and 300 °C. For the test at 200 °C, such a maximum stress would have been too close to the monotonic failure stress (Figure 8). A few cycles would have led to the failure of the specimen, which explains the choice of a lower maximum shear stress (500 MPa rather than 560 MPa).

Temperature (°C)	$\tau_{\min}$/$\tau_{\max}$ (MPa)	Number of Cycles	$\gamma_{\mathrm{final}}$ [−]	Final Stage
20	100/560	500, 600 (failure)	3.5	5
200	100/560	1 (monotonic case)	no data	no data
200	100/500	10,000	1.8	2
300	100/560	1000, 1400 (failure)	2.1	2

## Data Availability

Not applicable.

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
