# Peer review of "Development of Temperature-Controlled Shear Tests to Reproduce White-Etching-Layer Formation in Pearlitic Rail Steel"

_materials, 2022, doi:10.3390/ma15196590_

Round 1
Reviewer 1 Report
The article “Development of temperature-controlled shear-tests to reproduce white-etching-layer formation in pearlitic rail steel” is devoted to the formation of a “white etching layer” (WEL) on the rail surface during operation, which leads to an increased probability of defect formation.
In the paper, mechanical tests are carried out that simulate the formation of a WEL on the pearlitic rail steel. The paper contains the results of shear tests at various high temperatures and the results of the microstructure study of specimens after failure by optical and electron microscopy. It was found that under a certain thermo mechanical impact brittle zones are formed, which may be the centers of crack initiation in rail steel.
The paper is well structured. Among the strong suits of the article a good introduction with a wide reference list and the use of appropriate methods for microstructure research should be noted.
It is difficult to estimate the practical significance of the paper; however, the results obtained in this work, which confirm the formation of microstructure zones similar to a WEL during severe plastic shear strain, can be useful for more detailed further studies of the mechanism of the WEL formation on the rail surface during operation.
Some suggestions, questions and comments on the paper.
1. How correct is the expression “7.1 ± 7.5 µm” referring to the grain size (see Table 3)? It is clear that if we talk about the deviation of the value from the point of view of statistics, then it is correct. However, the grain size distribution law differs from the normal one, and also cannot take a negative value.
2. In table 2, I believe, a typo was made. It must be L = 4 mm.
3. To the discussion about the serrations of the diagrams shown in Figure 8. The Portevin-Le Chatelier effect can indeed manifest itself on the steel under study. However, taking into account the fact that for all diagrams the load oscillation amplitude is approximately the same, and the serrations appear for all diagram areas, including in the areas of elastic deformation and small plastic deformations, I believe that the authors are right about “experimental artifacts”. If the forces during loading of these samples were much less than the maximum load capacity of the testing machine, then this is due to fluctuations in the force sensor.
In general, the article leaves a positive impression and corresponds to the subject matter of the “Materials”.
Author Response
Please see the attachment. The answers of the questions are written in blue.

Reviewer 2 Report
In the manuscript “Development of temperature-controlled shear tests to reproduce white-etching-layer formation in pearlitic rail steel” authors performed monotonic and cyclic shear tests under controlled temperature using hat-shaped specimens and the Gleeble thermomechanical simulator in an attempt to reproduce the WEL formation induced at the rail surface.
The manuscript addresses an interesting topic; however, it needs to undergo an extensive review before being accepted for publication. Metrologically, the manuscript presents serious problems. The International System of Units was not attended. Measuring systems must be fully specified. Measurement procedures must be fully described so that other researchers can reproduce the results. The result traceability was not demonstrated and therefore the ISO/IEC 17025 standard was not met. The results were presented with different number of significant digits. Although the results were stated as a mean value and a range of variation, it was not stated what this range represents.
Some comments are presented in the attached file that may contribute to improving the manuscript quality.

Author Response
Please see the attachment. The answers of the questions are written in red.

Reviewer 3 Report
The article highlights peculiarities of formation of a white etching layer, a very hard and brittle phase on the rail surface. The authors performed monotonic and cyclic shear tests using hat-shaped specimens, in the temperature range of 20 °C to 400 °C to reproduce the white etching layer. The tests confirmed that the evolution stages leading to WEL formation could reflect a probabilistic criterion of crack initiation.
The article is interesting, but a number of shortcomings need to be corrected:
1. The authors should express their opinion on the prevailing influence of the load applied to the rail (Mode I or Mode II) on the peculiarities of the change in the microstructure and properties of steel. Of course, each type of load (Mode I or Mode II) will have its own characteristics, but it is necessary to express an opinion about which type of load is more dangerous.
2. It is necessary to compare the trends of changes in the microstructure and mechanical characteristics for Mode II with the results obtained for Mode I, since during operation the rail is subjected to the force of Mode I and Mode II. For the load at Mode I, the authors can use the data given in particular in https://doi.org/10.1007/s11003-013-9539-9.
3. It would be good to present the results of the evaluation of the residual stresses in the structural components of the near-surface layers of steel, which undergoes changes under the influence of thermo-power factors.
4. It is necessary to evaluate the microfractographic features of fractures obtained for the studied steel under cyclic loading and different heating temperatures. There are very few results of such studies in the literature.
5. The article must be formatted according to the journal requirements.
Author Response
Please see the attachment. The answers of the questions are written in brown.

Round 2
Reviewer 2 Report
The authors have satisfactorily addressed my comments. The article may be accepted for publication in its current form.
Two comments are presented below.
1) Remember that the zeros to the right of the last non-zero digit are significant digits. Thus, in Table 3 the Aspect ratio is equal to 2.0 ± 0.9. In Tables 5 and 7, the Aspect ratio is equal to 2.0 ± 1.0
2) About the answer to 7. On Page 7, about results.
a) Comment if it was possible to materialize exactly these temperatures (20 °C, 200 °C, 300 °C, and 400
°C) or if there was a variation around these values.
Small temperature variations (a few degrees) are observed during mechanical loading. These temperature variations are negligible compared to the high temperatures applied.
Although this temperature variation is negligible compared to the high temperatures applied, it must be declared.
Reviewer 3 Report
The authors took into account all comments of the reviewer and made appropriate corrections to the manuscript.